# White Sweet Potato as Meal Replacement for Overweight White-Collar Workers: A Randomized Controlled Trial

**DOI:** 10.3390/nu11010165

**Published:** 2019-01-14

**Authors:** Chun-Kuang Shih, Chiao-Ming Chen, Tun-Jen Hsiao, Ching-Wen Liu, Sing-Chung Li

**Affiliations:** 1School of Nutrition and Health Sciences, College of Nutrition, Taipei Medical University, 250 Wu-Hsing Street, Taipei 11031, Taiwan; ckshih@tmu.edu.tw; 2Department of Food Science, Nutrition, and Nutraceutical Biotechnology, Shih Chien University, No. 70, Dazhi St., Zhongshan Dist, Taipei 10462, Taiwan; charming@g2.usc.edu.tw; 3Chinese Taipei Society for the Study of Obesity, 250 Wu-Hsing Street, Taipei 11031, Taiwan; antifat53@gmail.com; 4Department of Food Science, Tunghai University, No. 1727, Sec. 4, Taiwan Boulevard, Taichung 40704, Taiwan; stevenliu@thu.edu.tw

**Keywords:** white sweet potato, meal replacement, overweight, glycated hemoglobin

## Abstract

Overweight and obesity are a global concern. Meal replacements (MRs) are portion- and calorie-controlled meals, which make the food environment part of an individual’s weight loss regimen. White sweet potato (WSP; *Ipomoea batatas* L.), used in traditional medicine in Brazil, Japan, and Taiwan, is a healthy carbohydrate source. In this randomized controlled trial, we assessed the effects of a WSP formula on body weight management in 58 white-collar workers through MR to elucidate the effects of this WSP-MR on factors leading to overweight. The participants consumed either two packs a day for a total of 132 g of WSP (WSP-MR group) or a normal diet daily (non-WSP group) for eight weeks. After eight weeks, body weight, body fat, body mass index, wrist circumference, thigh circumference, calf circumference, mid-arm circumference, and triceps skinfolds decreased significantly in both the groups. Moreover, the WSP-MR group demonstrated a 5% decrease in body weight, body fat, body mass index, and mid-arm circumference and a 3.5% decrease in glycated hemoglobin levels (*p* < 0.05). The treatment was well tolerated, without side effects or adverse events. Thus, our WSP formula as an MR can facilitate individual weight loss and thus has commercial application in the food industry.

## 1. Introduction

Overweight and obesity has rapidly become a leading public health concern in many countries. Nutrition and health surveys found that the overweight and obesity prevalence was 37.1% in China in 2002 [1] and 43.3% in Taiwan during 2005–2008 [2]. In the United States, the obesity prevalence among workers with relatively low-obesity occupations (i.e., white-collar jobs) significantly increased between 2004–2007 and 2008–2011, but that among workers with high-obesity occupations (i.e., blue-collar jobs) did not [3]. Therefore, provision of nutritious, healthy, convenient, and safe meal replacement (MR) foods to the overweight or obese population has become a topic of research interest.

A person may substitute a regular meal with an MR for several reasons. MRs are portion- and calorie-controlled meals, which aid in making individuals’ food environments more selective for long-term weight loss [4]. Liquid MR shakes in combination with an energy-restricted diet for weight loss maintenance can more efficiently facilitate weight reduction compared with dieting alone [5,6]. However, MRs have limitations; for instance, insufficient information about MRs before beginning the diet can lead to nutritional imbalance. A meta-analysis indicated that a partial MR plan had greater weight loss efficacy than did a conventional reduced calorie diet, despite equivalent calorie goals for the groups [7]. Thus, the use of MRs may help in reducing the caloric intake by not only reducing the assigned caloric intake level but also minimizing contact with and preparation of foods during nutritional interventions on weight maintenance [8].

White sweet potato (WSP; *Ipomoea batatas* L.) of the Convolvulaceae family is used as a healthy source of carbohydrate and is part of traditional medicine in Brazil, Japan, and Taiwan [9]. In patients with type 2 diabetes mellitus (T2DM), the tuberous WSP root can reduce insulin resistance as well as fasting plasma glucose, low-density lipoprotein-cholesterol (LDL-C), and fibrinogen levels and increase adiponectin levels [10,11,12]. Ju et al. reported that a 30% dietary supplementation of sweet potato or purple sweet potato considerably ameliorated high-fat diet (HFD)–induced obesity in mice, as well as reducing their body weight and fat accumulation and improving their lipid profile and energy expenditure modulation [13]. Plant-based MRs, such as soy protein formula, can be an effective treatment for weight loss and fat mass reduction in obese people with or without T2DM [14,15].

Maltodextrins (MDs) are polysaccharide produced from starch through partial enzymatic or acid hydrolysis, followed by purification and spray drying. Because of their varied applications, MDs are ubiquitous in foods, beverages, and dietary supplements. However, because glucose from digested MDs is rapidly absorbed in the small intestine, the increased MD use has had potential effects on metabolism and health [16]. In recent years, MDs have been the main resource of carbohydrate in MRs; however, the increased use of refined carbohydrates, such as isolated starches and MDs, can increase obesity rates [17]. Thus far, clinical studies exploring functional tubers as an ingredient to replace MDs in MR formulas have been scant. Therefore, our current experimental hypothesis is that a sweet potato–replaced food product, developed according to the recommended dietary allowance, provides the balanced nutritional requirement for weight loss. Thus, in this study, we assessed the effects of a WSP-MR formula on body weight management among white-collar workers by using an MR strategy in a dietary intervention trial.

## 2. Participants and Methods

### 2.1. Study Participants

Eligible white-collar workers, aged 30–50 years and overweight according to the body mass index (BMI) range of 24–30, according to the Department of Health (DOH) in Taiwan, in the regular health check-ups were recruited from Chunghwa Picture Tubes, Ltd. (Taoyuan, Taiwan). Of the 80 employees informed about this study, 72 individuals agreed to participate and to using their health check-up data. Exclusion criteria were being currently enrolled in another organized weight loss program, lactose intolerance, taking medications that affect appetite, history of gastric bypass or other surgical weight loss procedures, medical conditions (e.g., cancer, substance abuse, and psychiatric disorders), taking vitamins or mineral supplements, unable to join nutrition courses, and contraindications to exercise and changing eating habits. Finally, after a detailed explanation of the study, 60 participants were qualified to participate.

All participants provided their informed written consent; basic anthropometric measurements, such as body height, body weight, BMI, waist circumference (WC), hip circumference (HC), wrist circumference (WrC), waist-hip (W-H) ratio, thigh circumference (ThC), calf circumference (CaC), mid-arm circumference (MAC), mid-arm muscle circumference (MAMC), triceps skinfolds (TrS), blood pressure (BP), and pulse rate, were determined. Before participating in the study, the participants were offered the opportunity to discuss any queries with the primary investigator, physician, and study coordinator. The flow of participant selection is illustrated in Figure 1. The Institutional Review Board of Taipei Medical University approved the study protocol (TMU-JIRB N201604045); this trial is registered at ClinicalTrials.gov (NCT02935179).

### 2.2. Trial Design

In this open-label, randomized, parallel dietary intervention trial, participants were randomly assigned to the non-WSP or WSP-MR groups by using computer-generated identification numbers based on the order of recruitment. The anthropometric measurements, blood samples, dietary records, and clinical and biochemical data were collected at baseline and the end of post intervention week eight. Women and men in both the groups were instructed to follow an energy-restricted diet of approximately 1200 and 1500 kcal per day, respectively. The normal diet in daily three meals comprised vegetables, fruits, whole grain cereals, lean meat, and low-fat dairy products. A registered dietitian suggested and validated the total energy intake and nutritional assessment of three major nutrients according to the percentile: 55% carbohydrate, 30% fat, and 15% protein.

The WSP-MR group was advised to replace two daily meals, namely lunch and dinner, with two packs of shakes and one normal diet meal. The participants received a daily serving of two packets with 132 g of WSP-MR formula (21 g of WSP from 66-g packets). In each packet, 400 mL of warm water was added to provide heat density of 0.65 kcal/mL for each meal. The calorie values and nutrient contents of WSP and WSP-MR are listed in Table 1. The WSP-MR formula represents a balanced diet product for weight reduction according to dietary reference intakes (DRIs) provided by the Ministry of Health and Welfare, Taiwan. The participants were provided sufficient MR sachets free of charge; their compliance was assessed by counting the number of leftover sachets every two weeks.

During an 8-week intervention period, all participants attended four group-training sessions and four face-to-face visits for nutrition education. All participants received instruction manuals including a sample meal plan, recipes, and information regarding physical activities. Group-training sessions were held separately for each group by a nutrition advisor. The nutritional session topics included practical knowledge of food, portion sizes, and meal frequency, all with the aim of changing dietary behavior and lifestyle. At baseline, the participants were instructed on food selection, portion-size estimation, and accurate dietary-intake recording. To ensure compliance with their diet and physical activity during the trial, the participants were asked to record their food intake and physical activities during the weekly intervention period. Dietary records were reviewed for each study participant and analyzed using nutrition analysis and statistical software [18]. WC, HC, WrC, ThC, Ca, MAC, MAMC, and TrS were determined using a flexible tape and electronic skinfold caliper (Skyndex, NM, USA). BP was measured under standardized conditions after a 10-min resting period. Anthropometric measurements and venous blood sampling were performed in the morning after an overnight fasting of at least 12 h at baseline and after the 8-week intervention.

### 2.3. WSP-MR Formula

The special sweet potato cultivar WSP (Tainung 10, TNG10, Taiwan) was cultivated by an agricultural institution in Taiwan. The carbohydrate content in the WSP-MR was replaced with 30% (by weight) of TNG10; moreover, additional proteins, fats, vitamins, and minerals were added make it a balanced diet. In the WSP formula, carbohydrate, protein, and fat contributed 52%, 23%, and 25% cal, respectively. A 100 g of WSP-MR contains 390.7 kcal, with 22.4, 10.9, 58.5, and 15.5 g of protein (AOAC 990.03 method), fat (CNS5036 method), carbohydrate, and dietary fiber (AOAC 985.29 method), respectively. This ingredient analysis was validated by the Food Industry Research and Development Institute of Taiwan. To facilitate adherence, all WSP-MRs were provided by Han-Sient Trading Co. (New Taipei City, Taiwan). and given to the participants free of charge.

### 2.4. Biochemical Analyses

Blood was obtained through capillary blood collection; fasting blood glucose (FBG) was examined instantly using the One-Touch FreeStyle Freedom Lite blood glucose monitor (Abbott Diabetes Car Inc., CA, USA). Blood biomarkers, such as insulin, triglycerides, total cholesterol (TC), aspartate transaminase (AST), alanine transaminase (ALT), creatinine, blood urea nitrogen (BUN), uric acid, high-sensitivity C-reactive protein (hs-CRP), and LDL-C, and high-density lipoprotein-cholesterol (HDL-C), were analyzed from a 4-mL blood sample by using the Beckman Synchron LX-20 (Beckman Coulter Inc., CA, USA). Glycated hemoglobin (HbA1c) levels were analyzed from another 1.5-mL blood sample by using Spotchem SP-4410 (Arkray Inc., Kyoto, Japan). All the listed experiments and procedures were performed in the authorized Yea-Tong laboratory center, in Chung-Li City, Taiwan.

### 2.5. Statistical Analyses

Statistical analyses were performed using SPSS (version 19; SPSS Inc., Chicago, IL, USA). For intragroup testing, we performed the paired *t* test with the Wilcoxon signed-rank test to compare the differences in participants’ anthropometric measurements, glycemic markers, and lipid profiles in each group. For intergroup comparisons, we performed the independent *t* test with the Mann–Whitney *U* test to compare the final values or nets between the non-WSP and WSP-MR groups. All data are expressed as means ± standard deviations (SDs). The results were considered statistically significant for *p* < 0.05.

## 3. Results

### 3.1. Participant Characteristics

We recruited 80 participants at baseline; however, for personal reasons and after professional evaluation by physicians and clinical nutritionists, only 56 overweight participants were eligible for per protocol analysis (Figure 1). Four participants dropped out: three non-WSP participants due to personal reasons and one WSP-MR participant because they disliked the MR’s flavor. All participants tolerated MRs well. The overall response rate was 93.33% (90% and 96.67% in the non-WSP and WSP-MR groups, respectively). The baseline demographic (age and sex), anthropometric (body height, body weight, BMI, body fat, WC, HC, W-H ratio, and basal metabolic rate), and clinical characteristics (BP, FBG, insulin, TC, TG, HDL-C, LDL-C, uric acid, AST, and ALT) of the patients in both groups were comparable (Table 2). The average age of white-collar workers was 37.33 ± 5.5 in the non-WSP group and 38.76 ± 6.24 in the WSP-MR group. The BMI was 24.68 ± 0.9 in the non-WSP group and 24.99 ± 1.05 in the WSP-MR group. No statistically significant difference for any variable of comparison was detected; no difference appeared between the groups in age and sex. In addition, equality between the groups were assumed in terms of the anthropometric measurement of body height, body weight, BMI, body fat, WC, HC, W-H ratio, and basal metabolic rate. Furthermore, no significant differences were found when comparing the groups in terms of clinical variables, namely BP, FBG, insulin, TC, TG, HDL-C, LDL-C, uric acid, AST, and ALT. According to participants’ food diaries, calorie, carbohydrate, protein, fat, and dietary fiber intake did not differ significantly between the two groups.

### 3.2. Changes in Anthropometric Parameters and Clinical Characteristics

At baseline, the energy intake was similar in both the groups. After the 8-week intervention, energy intake significantly decreased within each group, but not between both groups. Table 3 presents the changes in patients’ clinical characteristics between the endpoint (week 8) and baseline (week 0) in the groups. Dietary intervention significantly reduced body weight, body fat, BMI, WrC, ThC, CaC, MAC, and TrS in both the groups; however, only the changes in body weight, body fat, BMI, and MAC were significant. The relative changes in body weight (−2.11 ± 1.70 vs. −3.70 ± 2.24 kg), body fat (−0.95% ± 2.21% vs. −2.26% ± 2.09 %), BMI (−0.76 ± 0.68 vs. −1.33 ± 0.82 kg/m^2^), and MAC (−0.85 ± 0.74 vs. −1.49 ± 0.96 cm) were higher in the WSP-MR group than in the non-WSP group (*p* < 0.05). After the 8-week intervention, the within-group analysis revealed a significant 5% decrease in body weight, body fat, BMI, and MAC in the WSP-MR group.

### 3.3. Changes in Biochemical Characteristics, Glycemic Markers, and Lipid Profiles

The relative changes in albumin (−0.23 ± 0.19 vs. 0.04 ± 0.15 g/dL), phosphorus (−0.04 ± 0.37 vs. 0.16 ± 0.35 meq/L), and BUN (−1.59 ± 2.22 vs. 0.14 ± 3.07 mg/dL) levels were significantly higher in the WSP-MR group than in the non-WSP group. The decline in serum calcium concentration (−0.69 ± 0.25 vs. −0.36 ± 0.20 meq/L) was significantly lower in the WSP-MR group than in the non-WSP group. The changes in the levels of liver and renal function markers, such as AST, ALT, γ-GTP, total bilirubin, and uric acid, did not demonstrated significant within- and between-group differences (Table 4). The non-WSP and WSP-MR groups both demonstrated nonsignificant changes in FBG levels after dietary intervention (−6.52 ± 5.6 and −6.38 ± 5.3 mg/dL, respectively). Notably, the changes in HbA1c levels were significantly lower in the WSP-MR group than in the non-WSP group (−0.02% ± 0.17% vs −0.19% ± 0.20%). After the 8-week intervention, the non-WSP and WSP-MR groups both demonstrated significant changes in the levels of lipid markers such as TC (−13.59 ± 12.91 and −10.83 ± 18.83 mg/dL, respectively), TG (−26.3 ± 32.11 and −33.17 ± 47.23 mg/dL, respectively), and LDL-C (−12.33 ± 13.08 and −10.07 ± 17.51 mg/dL, respectively). By contrast, the changes in the HDL-C levels were significantly higher within in the WSP-MR group (49.07 ± 7.77 vs. 51.52 ± 8.5 mg/dL, *p* < 0.001); however, no such significant between-group change was noted (Table 5).

## 4. Discussion

Our current results indicated that both our weight loss strategies were effective for treating overweight white-collar workers. Our well-developed MR developed using WSP reduced body weight, body fat, BMI, and MAC by 5% and HbA1c by 3.5%. In a systematic review of randomized controlled trials using partial MR, such meal interventions were found to safely and effectively cause weight loss and improve weight-related risk factors for disease [7]. Although no significant intergroup or intragroup differences were noted in WC and HC, the 8-week WSP-MR intervention significantly reduced WC and HC. WC can predict body fat mass [19], obesity-related metabolic index [20], and cardiovascular risk factors [7]. In addition to WC, using skinfold caliper is a good alternative to dual-energy X-ray absorptiometry or computed tomography when predicting the amount of fat in the central and peripheral bodies [19]. Although no between-group difference in the reduction in TrS was noted, the groups exhibited decreases of 22.5% and 23.2% compared with the baseline (Table 3).

MRs are portion- and calorie-controlled meals, which make the food environment part of an individual’s weight loss regimen, especially busy office workers and people eat outside. A challenge for the behavioral treatment of obesity is achieving durable improvements in various self-control skills such as making individuals’ food environments more compatible with long-term weight loss. Lifestyle change treatments for weight loss produce medically meaningful weight reductions, but lost weight is usually regained. MRs represent one avenue for improving long-term weight loss [4]. Weight loss can be achieved using a conventional, structured, energy-restricted modified diet alone or in combination with MRs for weight control [21]. Gulati et al. demonstrated that high-protein MRs could lead to significant weight loss and alleviate overweight and obese in North Indians [22]. High-protein MRs led to greater mean weight loss and percent BMI reduction than did standard-protein MRs after 12 weeks; however, the difference between overweight and obese Chinese people with hyperlipidemia was nonsignificant [23]. High-protein and low-carbohydrate diets cause greater weight loss in the short-term than do diets emphasizing overall energy restriction; however, at the end of one year, both diets demonstrate similar results [24]. From the nutritional perspective, a low-calorie MR must contain a complete range of nutrients, fiber, minerals, and trace elements, which can replace the stable food and comply with specifications, such as low fat levels (20%–25% of total calories), high dietary fiber content (5 g of fiber/200 cal), nutrient or dietary ingredient saturation, and delay in gastrointestinal digestion and absorption. Due to similar calorie diets, exercise recommendations consist of three major nutrients ratios (55% carbohydrate, 30% fat, and 15% protein) between the two groups. The WSP-MR could provide 15.5 g of fiber and the average glycemic index was 36.2, which may cause slow down the energy absorption of the subject. The Low-GI foods may enhance weight control because they promote satiety, minimize postprandial insulin secretion, and maintain insulin sensitivity. This hypothesis is supported by several intervention studies in humans in which energy-restricted diets based on low-GI foods produced greater weight loss than did equivalent diets based on high-GI foods [25].

Sweet potatoes are a better source of carbohydrates, various vitamins, minerals, and protein than are other vegetables [26]. To our knowledge, this is the first study to report WSP use in MR formulas applied to overweight white-collar workers. In our previous study, five groups of HFD-induced obesity mice were administered a normal diet, HFD, HFD + 12% WSP, HFD + 24% WSP, and HFD + 36% WSP for four weeks. Both the HFD + 24% WSP and HFD + 36% WSP groups had lower weight than did the HFD group (unpublished data). Therefore, in the current study, we used a MR formula with 30% of WSP in WSP-MR to overweight participants, a proportion similar to that used by Ju et al. [13]. A decrease in ‘whole’ food and dietary fiber consumption, along with an increase in the consumption of rapidly digestible and absorbable carbohydrate sources, such as isolated starches, starch derivatives, and sugars, increased the global prevalence of obesity, diabetes, and cardiovascular disease [17]. MD is traditionally used to replace the carbohydrates source in MRs and increase a person’s risk of having high cholesterol levels, weight gain, and T2DM. Our current data suggests that overweight participants receiving two packs per day of WSP-MR (a total of 132 g) not only provides balanced nutrients but also reduces HbA1c and increases HDL cholesterol levels (Table 5). The decreases in body weight, waist, and HbA1c notes in the study are similar to those noted in a previous study using two partial MRs, which provided further management options for overweight or obese individuals with T2DM [27].

By contrast, traditional dietary recommendations for weight loss endorse a fat- and calorie-restricted diet, high in complex carbohydrates. Nevertheless, the clinical practice guidelines do not provide additional recommendations to prevent or control overweight. Studies examining the ideal amount of carbohydrate intake for overweight people have reported inconclusive results. Our recipe featured WSP, pea protein, and plant oil as carbohydrate, protein, and fat sources, respectively, along with a low glycemic index and the following energy distribution: 52% carbohydrate, 22.9% protein, and 25.1% lipid. In our preclinical trial using oral glucose tolerance tests in 20 healthy participants, we noted that the average glycemic index was 36.2 for WSP-MR. The consensus of nutritional recommendations for normal diet by the American Heart Association, American Cancer Society, American Dietetic Association, American Academy of Pediatrics, and the National Institutes of Health suggests consuming 55% carbohydrate, 30% fat, and 15% protein [24]. However, our participants received a low-calorie diet and a modest increase in protein content and glycemic index; this led to an improved rate of completion of the intervention and maintenance of weight loss [28]. Moreover, weight loss diets selected by patients generally have the targeted percentages of energy distribution {fat (20%–40%), protein (15%–25%), and carbohydrates (35%–65%)} without significant differences in the dietary composition [29].

In a review, Howarth et al. indicated that when energy intake is *ad libitum*, mean additional consumption of fiber at 14 g/day is associated with a 10% decrease in energy intake and 1.9 kg of body weight loss over 3.8 months [30]. According to a recent survey of nutritional intake in Taiwan, adults consumed 16.7 g of dietary fiber daily [31]. In our study, the daily intake of WSP-MR could provide 15.5 g fiber, reaching 51% DRI through the two meals. Because of low dietary calories and high dietary fiber intake, body weight loss can theoretically be improved.

Resistant starch (RS) may be a promising dietary fiber for the prevention or treatment of obesity and its related diseases [32]. According to the source and processing procedure, RS is currently classified in five categories: RS1, RS2, RS3, RS4 and RS5. Consumption of food containing RS4-enriched flour (30% *v*/*v* RS4) for 2–12 weeks significantly reduced TC and non-HDL-C levels by 5.5% and 12.8%, respectively, in 89 patients with metabolic syndrome compared with regular or control flour food [33]. In our previous WSP digestibility test, the RS and slowly digestible starch (SDS) contents in WSP were 0.36 ± 0.05 and 48.0 ± 1.8 mg/100 mg, respectively. Therefore, whether the higher SDS content in WSP contributes to weight loss in obese patients warrants further research.

In a study of polyphenols and phenolic acids in sweet potato roots, phenolic compounds contents were determined in raw peeled roots, jackets of raw roots, and water-steamed sweet potato roots. Total polyphenol content ranged from 1161 (O’Henry, flesh-raw) to 13,998 (414, peel-raw) mg/kg, caffeic acid content ranged from undetected (414, flesh-raw) to 320.7 (Beauregard, peel-raw) mg/kg, and 3-caffeoylquinic acid content ranged from 57.57 (O’Henry, flesh-raw) to 2392 (414, peel-raw) mg/kg [34]. Yoshino et al. reported that after 12 weeks of supplementation with resveratrol (75 mg/day), there was no decrease in body weight for non-obese women with normal glucose tolerance [35]. A phytochemical and mineral-rich filtered sugarcane molasses concentrate (FMC) as an agent that can reduce insulin responses and lessen the load on the pancreatic beta cells [36]. However, all phenolic compounds responsible for the differences in the results differences warrant additional studies involving urinalysis.

Both the non-WSP and WSP-MR groups demonstrated good dietary compliance and effective weight loss. However, this study has limitations. First, the small sample size and short trial duration may have failed to detect significant changes in the study measures. Second, some participants did not supply complete data regarding dietary compliance in their food diaries at baseline or at the endpoint. Third, both groups had similar exercise recommendations; thus, the effects of physical activity and weight loss could not be assessed separately. In addition, to confirm the differences between the two groups, biochemical factors, such as SDS in the gut and phenolic compounds in urine, require analysis. The strengths of this study are that it represents WSP use in MR formulas applied to overweight white-collar workers. The low GI of WSP-MRs are also a convenient and feasible MR for people who require weight maintenance.

## 5. Conclusions

The use of our WSP-MR formula effectively reduced body weight, body fat, BMI, and MAC, and HbA1c levels. This treatment was well tolerated, and no serious side effects were detected. Use of plant tubers as a carbohydrate source in regular diet can provide not only energy but also other nutrients that prevent malnutrition (e.g., fiber, vitamins, and minerals) in individuals requiring weight loss.

## Figures and Tables

**Figure 1 nutrients-11-00165-f001:**
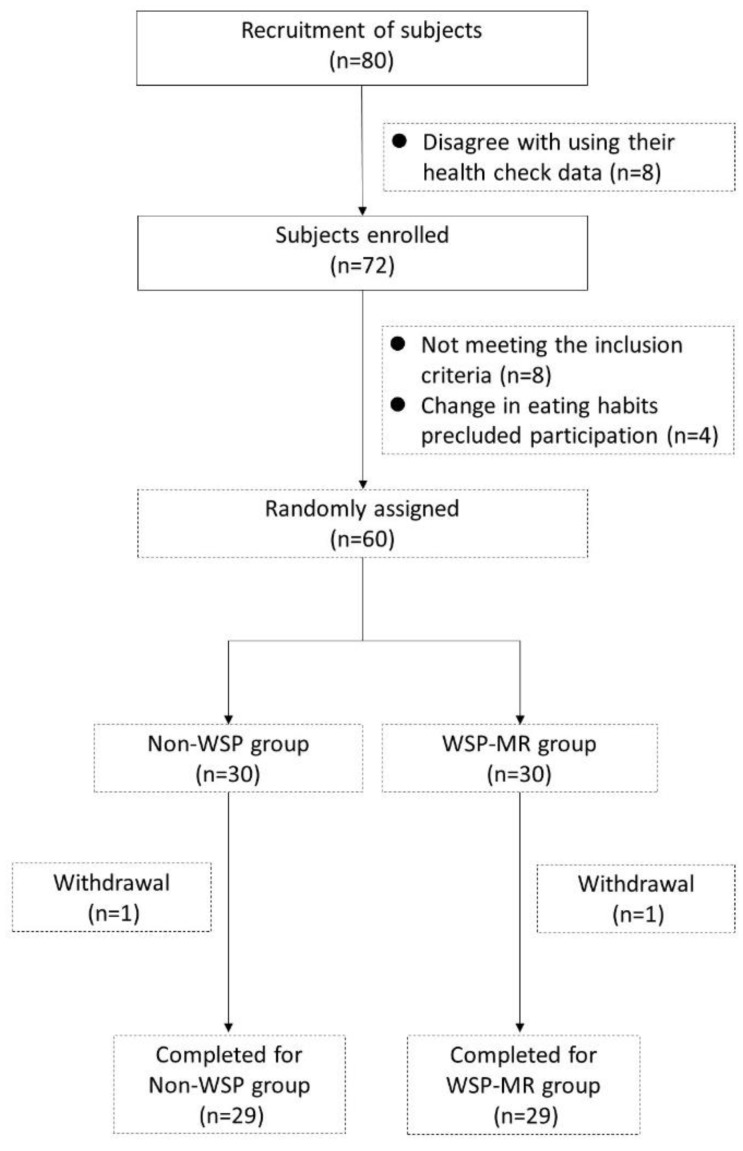
Flowchart of participant selection.

**Table 1 nutrients-11-00165-t001:** Nutrient composition of white sweet potato and white sweet potato-meal replacement per 100 g.

Nutrient	WSP	WSP-MR
Calories (kcal)	380.8	390.7
Protein (g)	2.9	22.4
Total fat (g)	0	10.9
Saturated Fat	0	2.5
MUFA	0	4.9
PUFA	0	2.9
Carbohydrates (g)	92.3	58.5
Dietary fiber	10.1	15.5
Sugars	31.1	13.4
Water (g)	9.6	3.8
Ash (g)	3.0	4.5
Minerals		
Sodium (mg)	176	410.6
Manganese (mg)	3.34	1
Potassium (mg)	11,700	469
Calcium (mg)	1080	610.3
Magnesium (mg)	675	223.5
Iron (mg)	18.3	6.4
Zinc (mg)	7.98	7.3
Phosphate (mg)	2660	317.5
Chloride (mg)	2500	356.1
Copper (ug)	5.38	364

WSP: White sweet potato; WSP-MR: White sweet potato meal replacement.

**Table 2 nutrients-11-00165-t002:** Demographic and clinical characteristics of the participants.

Characteristic	Non-WSP	WSP-MR	*p*
Age (year-old)	37.33 ± 5.50	38.76 ± 6.24	0.371
Male/female (*n*)	13/14	14/15	0.992
Height (cm)	163.91 ± 9.17	164.58 ± 7.81	0.767
Body weight (kg)	66.77 ± 8.4	68.01 ± 6.88	0.546
Body mass index (kg/m^2^)	24.68 ± 0.90	24.99 ± 1.05	0.245
Body fat (%)	29.12 ± 5.39	29.42 ± 6.37	0.851
WC (cm)	84.49 ± 7.67	85.87 ± 4.78	0.427
HC (cm)	99.24 ± 10.51	101.87 ± 3.39	0.206
W-H ratio (%)	86.52 ± 16.24	84.33 ± 4.49	0.487
Basal metabolic rate (kcal)	1507.16 ± 203.52	1508.40 ± 176.10	0.981
Systolic blood pressure (mmHg)	116.59 ± 12.24	112.45 ± 15.01	0.265
Diastolic blood pressure (mmHg)	69.41 ± 9.60	68.31 ± 15.63	0.751
FBG (mg/dL)	86.81 ± 4.69	85.28 ± 7.33	0.357
Insulin (uU/mL)	9.41 ± 4.86	8.63 ± 3.53	0.493
TC (mg/dL)	195.48 ± 26.68	188.24 ± 25.35	0.302
TG (mg/dL)	113.93 ± 41.91	111.31 ± 53.19	0.840
HDL-C (mg/dL)	50.44 ± 10.34	49.07 ± 7.77	0.574
LDL-C (mg/dL)	128.78 ± 24.08	123.14 ± 22.95	0.374
Uric acid (mg/dL)	5.33 ± 1.41	5.19 ± 1.28	0.691
AST (U/L)	29.96 ± 6.42	23.17 ± 23.84	0.800
ALT (U/L)	22.26 ± 9.53	29.17 ± 49.63	0.480

Data are expressed as means ± standard deviations analyzed by independent sample *t* or chi-square test. Non-WSP: normal diet daily; WSP-MR: white sweet potato meal replacement; WC: waist circumference; HC: hip circumference; W-H ratio: waist-hip ratio; FBG: fasting blood glucose; AST: aspartate transaminase; ALT: alanine transaminase.

**Table 3 nutrients-11-00165-t003:** Changes in participants’ anthropometric parameters over weeks 0–8.

	Non-WSP			WSP-MR		
	Week 0	Week 8	Changes	Week 0	Week 8	Changes
Body weight (kg)	66.77 ± 8.40	64.66 ± 8.39 *	−2.11 ± 1.70	68.01 ± 6.88	64.31 ± 6.21 *	−3.70 ± 2.24 ^#^
Body fat (%)	29.12 ± 5.39	28.17 ± 5.31 *	−0.95 ± 2.21	29.42 ± 6.37	27.16 ± 6.07 *	−2.26 ± 2.09 ^#^
BMI (kg/m^2^)	24.68 ± 0.90	23.92 ± 0.96 *	−0.76 ± 0.68	24.99 ± 1.05	23.65 ± 1.22 *	−1.33 ± 0.82 ^#^
WC (cm)	84.49 ± 7.67	82.93 ± 6.11	−1.57 ± 4.42	85.87 ± 4.78	83.78 ± 5.65 *	−2.10 ± 4.46
HC (cm)	99.24 ± 10.51	99.22 ± 3.52	−0.02 ± 8.91	101.87 ± 3.39	98.98 ± 3.31 *	−2.89 ± 2.88
W-H ratio (%)	86.52 ± 16.24	83.54 ± 4.80	−2.99 ± 15.99	84.33 ± 4.49	84.64 ± 5.14	0.32 ± 3.88
WrC (cm)	15.80 ± 1.01	15.59 ± 0.88 *	−0.21 ± 0.44	16.00 ± 1.04	15.58 ± 1.04 *	−0.43 ± 0.71
ThC (cm)	59.46 ± 2.35	57.29 ± 2.35 *	−2.18 ± 2.82	59.90 ± 3.11	57.15 ± 2.52 *	−2.76 ± 3.32
CaC (cm)	37.37 ± 1.84	37.00 ± 1.93 *	−0.37 ± 0.82	37.97 ± 1.69	37.33 ± 1.58 *	−0.64 ± 0.88
MAC (cm)	29.69 ± 2.31	28.84 ± 2.15 *	−0.85 ± 0.74	29.96 ± 1.93	28.47 ± 1.82 *	−1.49 ± 0.96 ^#^
MAMC (cm)	22.61 ± 2.61	23.36 ± 3.10 *	0.74 ± 1.89	22.76 ± 2.79	22.94 ± 2.50	0.18 ± 2.02
TrS (mm)	22.54 ± 4.79	17.47 ± 4.62 *	−5.07 ± 5.58	22.92 ± 5.84	17.60 ± 4.28 *	−5.31 ± 5.65

Data are presented as means ± standard deviations. * Significant within-group difference after intervention (paired *t* test analysis; *p* < 0.05). # Significant between-groups difference after intervention (independent *t* test; *p* < 0.05). Non-WSP: normal diet daily; WSP-MR: white sweet potato meal replacement; WC: waist circumference; HC: hip circumference; W-H ratio: waist-hip ratio; WrC: wrist circumference; ThC: thigh circumference; CaC: calf circumference; MAC: mid-arm circumference; MAMC: mid-arm muscle circumference; TrS: triceps skinfolds.

**Table 4 nutrients-11-00165-t004:** Changes in participants’ biochemical characteristics over weeks 0–8.

	Non-WSP			WSP-MR		
	Week 0	Week 8	Changes	Week 0	Week 8	Changes
Albumin (g/dL)	4.83 ± 0.19	4.60 ± 0.24 *	−0.23 ± 0.19	4.51 ± 0.21	4.55 ± 0.24 *	0.04 ± 0.15 ^#^
Total protein (g/dL)	7.41 ± 0.31	7.29 ± 0.25 *	−0.13 ± 0.25	7.38 ± 0.40	7.33 ± 0.37 *	−0.05 ± 0.23
Sodium (meq/L)	137.30 ± 2.15	139.96 ± 1.23 *	2.66 ± 2.31	137.67 ± 1.24	139.60 ± 1.37 *	1.93 ± 1.21
Potassium (meq/L)	4.09 ± 0.30	4.18 ± 0.30 *	0.09 ± 0.29	4.16 ± 0.31	4.22 ± 0.29	0.06 ± 0.30
Calcium (meq/L)	10.02 ± 0.22	9.33 ± 0.24 *	−0.69 ± 0.25	9.67 ± 0.29	9.31 ± 0.31 *	−0.36 ± 0.20 ^#^
Phosphorus (meq/L)	3.21 ± 0.46	3.17 ± 0.42 *	−0.04 ± 0.37	3.14 ± 0.44	3.30 ± 0.33 *	0.16 ± 0.35 ^#^
Magnesium (meq/L)	2.05 ± 0.12	2.11 ± 0.13 *	0.06 ± 0.11	2.02 ± 0.11	2.14 ± 0.13 *	0.11 ± 0.11
AST (U/L)	21.96 ± 6.42	19.59 ± 6.40	−2.37 ± 6.23	23.17 ± 23.84	17.48 ± 3.86	−5.69 ± 22.14
ALT (U/L)	22.26 ± 9.53	21.78 ± 18.11	−0.48 ± 16.18	29.17 ± 49.63	16.31 ± 7.90	−12.86 ± 43.81
γ-GTP (U/L)	19.33 ± 10.04	16.44 ± 7.88 *	−2.89 ± 4.37	25.17 ± 35.13	13.72 ± 8.18	−11.45 ± 30.89
Total bilirubin (mg/dL)	0.72 ± 0.31	0.81 ± 0.31	0.09 ± 0.26	0.77 ± 0.34	0.78 ± 0.37	0.01 ± 0.29
Creatinine (mg/dL)	0.76 ± 0.14	0.68 ± 0.14 *	−0.09 ± 0.05	0.76 ± 0.18	0.68 ± 0.15 *	−0.09 ± 0.06
BUN (mg/dL)	13.26 ± 2.38	11.67 ± 2.60 *	−1.59 ± 2.22	11.79 ± 2.93	11.93 ± 2.94	0.14 ± 3.07 ^#^
Uric acid (mg/dL)	5.33 ± 1.41	5.14 ± 1.21 *	−0.19 ± 0.63	5.19 ± 1.28	5.16 ± 1.11	−0.03 ± 0.66
Hs-CRP (mg/dL)	0.21 ± 0.18	0.13 ± 0.05 *	−0.09 ± 0.17	0.27 ± 0.34	0.38 ± 0.88	0.11 ± 0.98

Data are presented as means ± standard deviations. * Significant within-group difference after intervention (paired *t* test analysis; *p* < 0.05). # Significant between-groups difference after intervention (independent *t* test; *p* < 0.05). Non-WSP: normal diet daily; WSP-MR: white sweet potato meal replacement; AST: aspartate transaminase; ALT: alanine transaminase; BUN: blood urea nitrogen; hs-CRP: high-sensitivity C-reactive protein.

**Table 5 nutrients-11-00165-t005:** Changes in participants’ glycemic markers and lipids profiles over weeks 0–8.

	Non-WSP			WSP-MR		
	Week 0	Week 8	Changes	Week 0	Week 8	Changes
FBG (mg/dL)	86.81 ± 4.69	80.30 ± 6.33 *	−6.52 ± 5.60	85.28 ± 7.33	78.90 ± 6.14 *	−6.38 ± 5.30
HbA1c (%)	5.14 ± 0.37	5.12 ± 0.44	−0.02 ± 0.17	5.38 ± 0.35	5.20 ± 0.25 *	−0.19 ± 0.20 ^#^
Insulin (uU/mL)	9.41 ± 4.86	8.04 ± 4.09	−1.37 ± 3.70	8.63 ± 3.53	5.71 ± 2.98 *	−2.92 ± 3.79
TC (mg/dL)	195.48 ± 26.68	181.89 ± 25.16 *	−13.59 ± 12.91	188.24 ± 25.35	177.41 ± 25.31*	−10.83 ± 18.83
TG (mg/dL)	113.93 ± 41.91	87.63 ± 40.23 *	−26.30 ± 32.11	111.31 ± 53.19	78.14 ± 25.49 *	−33.17 ± 47.23
HDL-C (mg/dL)	50.44 ± 10.34	51.78 ± 10.48	1.33 ± 3.81	49.07 ± 7.77	51.52 ± 8.50 *	2.45 ± 4.45
LDL-C (mg/dL)	128.78 ± 24.08	116.44 ± 23.43 *	−12.33 ±13.08	123.14 ± 22.95	113.07 ± 22.58 *	−10.07 ± 17.51

Data are presented as means ± standard deviations. * Significant within-group difference after intervention (paired *t* test analysis; *p* < 0.05). # Significant between-groups difference after intervention (independent *t* test; *p* < 0.05). Non-WSP: normal diet daily; WSP-MR: white sweet potato meal replacement; FBG: fasting blood glucose; HbA1c: Glycated hemoglobin.

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
