# Peer review of "White Sweet Potato as Meal Replacement for Overweight White-Collar Workers: A Randomized Controlled Trial"

_nutrients, 2019, doi:10.3390/nu11010165_

Round 1
Reviewer 1 Report
The authors are to be commended for conducting this study. This manuscript will be a nice contribution to the literature. There are, however, a few areas in need of further clarification.
The reason for the significant different in weight loss between the two groups is not entirely clear. If both groups were given similar calorie diets and exercise recommendations, what is the mechanism for the greater weight loss in the WSP group? There should to be more discussion about this.
2. Thee is a statement in the discussion that "sweet potatoes are the best vegetable" - this seems a bit too strong as there are many good vegetables that are high in fiber, nutrients, and good quality carbohydrates. This sentence should be revised to reflect this.
3. More information on the eligibility criteria for this study should be provided. If the goal was weight loss, why were non-overweight individuals included?
4. The strengths of this study should be described in the discussion.
Author Response
Reviewer 1
We greatly appreciate reviewer’s thoughtful comments that helped improve the manuscript.
Point 1: The reason for the significant different in weight loss between the two groups is not entirely clear. If both groups were given similar calorie diets and exercise recommendations, what is the mechanism for the greater weight loss in the WSP group? There should to be more discussion about this.
Response 1: In line 251, the revised statement in the discussion is “Due to similar calorie diets, exercise recommendations and three major nutrients ratios (55% carbohydrate, 30% fat, and 15% protein) between the two groups. The WSP-MR could provide 15.5 g fiber and average glycemic index was 36.2 that may causes slow down energy absorption in subject. The Low-GI foods may enhance weight control because they promote satiety, minimize postprandial insulin secretion, and maintain insulin sensitivity. This hypothesis is supported by several intervention studies in humans in which energy-restricted diets based on low-GI foods produced greater weight loss than did equivalent diets based on high-GI foods [1].
Point 2: Thee is a statement in the discussion that "sweet potatoes are the best vegetable" - this seems a bit too strong as there are many good vegetables that are high in fiber, nutrients, and good quality carbohydrates. This sentence should be revised to reflect this.
Response 2: In line 259, the revised statement in the discussion is “Sweet potatoes are a better source of carbohydrates, various vitamins, minerals, and protein”.
Point 3: More information on the eligibility criteria for this study should be provided. If the goal was weight loss, why were non-overweight individuals included?
Response 3: In line 75, the revised statement in the Participants and Methods is “Eligible white-collar workers, aged 30–50 years and overweight according to the body mass index (BMI) range of 24–30, according to the Department of Health (DOH) in Taiwan, in the regular health check-ups were recruited from Chunghwa Picture Tubes, Ltd. (Taoyuan)”. Taiwan adopted BMIs of 24 and 27 as the cutoffs for overweight and obesity, respectively [2]
Point 4: The strengths of this study should be described in the discussion.
Response 4: In line 324, the revised statement in the discussion is “The strengths of this study is that it represents WSP use in MR formulas applied to overweight white-collar workers. The low GI of WSP-MRs are also a convenient and feasible MR for people in weight maintenance”.
Reference:
1. Brand-Miller, J.C.; Holt, S.H.A.; Pawlak, D.B.; McMillan, J. Glycemic index and obesity. The American Journal of Clinical Nutrition 2002, 76, 281S-285S, doi:10.1093/ajcn/76/1.281S.
2. Pan, W.-H.; Flegal, K.M.; Chang, H.-Y.; Yeh, W.-T.; Yeh, C.-J.; Lee, W.-C. Body mass index and obesity-related metabolic disorders in Taiwanese and US whites and blacks: implications for definitions of overweight and obesity for Asians. The American journal of clinical nutrition 2004, 79, 31-39.
Reviewer 2 Report
This is an intersting study well described and well written. However, I do not understand why testing a supplementation at all. Why not try to learn people to eat the right things. There i s no discussion at all about the pedagogic challenge to learn people to have stable lifestyle habits. It is not a possible way to go to have people to eat powder instead of real foo containing important fibres and need of chewing. This must be discussed and weighed.
Author Response
Reviewer 2
We greatly appreciate reviewer’s thoughtful comments that helped improve the manuscript.
Point 1: This is an intersting study well described and well written. However, I do not understand why testing a supplementation at all. Why not try to learn people to eat the right things. There i s no discussion at all about the pedagogic challenge to learn people to have stable lifestyle habits. It is not a possible way to go to have people to eat powder instead of real foo containing important fibres and need of chewing. This must be discussed and weighed
Response 1: In line 234, the revised statement in the discussion is “MRs are portion- and calorie-controlled meals, which make the food environment part of an individual’s weight loss regimen, especially busy office workers and people eat outside. A challenge for the behavioral treatment of obesity is achieving durable improvements in various self-control skills such as making individuals’ food environments more compatible with long-term weight loss. Lifestyle change treatments for weight loss produce medically meaningful weight reductions, but lost weight is usually regained. MRs represent one avenue for improving long-term weight loss[1]”
References:
1. Lowe, M.R.; Butryn, M.L.; Zhang, F. Evaluation of meal replacements and a home food environment intervention for long-term weight loss: a randomized controlled trial. The American Journal of Clinical Nutrition 2018, 107, 12-19, doi:10.1093/ajcn/nqx005.
Reviewer 3 Report
The manuscript represents a significant contribution to the field. However, the manuscript can be improved with some suggestions below that the authors should consider
The authors should describe how the sample size was determined. Was the effect size considered and whether the use of a p-value of<0.05 was set before or after the study? It seems that analysis of variance (ANOVA) was not used? Why?
Have the authors determined the glycemic index (GI) of the test products (? It would be helpful to have this information and ideally included in the publication considering that there has been significant effects in glycemic parameters such as FBG, HbA1c and insulin. If not available and the authors cannot provide the information, this should be added in the discussion.
The manuscript cites the role of polyphenols and it might be worth citing references below where similar results, i.e. significant improvement of sub-optimal properties in sub-normal individuals (e.g. overweight), were observed
1. Yoshino, J.; Conte, C.; Fontana, L.; Mittendorfer, B.; Imai, S.; Schechtman, K.B.; Gu, C.; Kunz, I.; Rossi Fanelli, F.; Patterson, B.W.; Klein, S. Resveratrol supplementation does not improve metabolic function in nonobese women with normal glucose tolerance. Cell Metab. 2012, 16, 658-64, doi: 10.1016/jNaNet.2012.09.015.
Ellis, T.P.; Wright, A.G.; Clifton, P.M.; Ilag, L.L. Postprandial insulin and glucose levels are reduced in healthy subjects when a standardised breakfast meal is supplemented with a filtered sugarcane molasses concentrate. Eur J Nutr. 2016, 55, 2365-2376, doi:10.1007/s00394-015-1043-6.
Minor edits
In the abstract, it stated that there were 56 subjects but the results section shows a total of 58.
Table 5 - formatting of the data in the first 3 rows (FBG, HbA1c and insulin) - alignment of rows
Author Response
Reviewer 3
We greatly appreciate reviewer’s thoughtful comments that helped improve the manuscript.
The manuscript represents a significant contribution to the field. However, the manuscript can be improved with some suggestions below that the authors should consider
Point 1: The authors should describe how the sample size was determined. Was the effect size considered and whether the use of a p-value of<0.05 was set before or after the study? It seems that analysis of variance (ANOVA) was not used? Why?
Response 1: We explain the process of sample size calculation on the basis of articles retrieved by a selective search of the international literature, as well as our own experience. Sample size calculation requires the collaboration of experienced biostatisticians and physician-researchers[1]. Botht co-PIs who are physician and dietion (Tun-Jen Hsiao, MD, PhD; Chiao-Ming Chen, PhD ) contribute medical knowledge ample size calculation in this clinical trials. We use of a p-value of<0.05 was set after the study. We used the paired t test to compare the differences between the two groups (Non-WSP vs WSP-MR). If more than two groups are used, we must use Anova as a tool to analyze.
Point 2: Have the authors determined the glycemic index (GI) of the test products (? It would be helpful to have this information and ideally included in the publication considering that there has been significant effects in glycemic parameters such as FBG, HbA1c and insulin. If not available and the authors cannot provide the information, this should be added in the discussion.
Response 2: The average glycemic index of WSP-MR was 36.2 (line 254). In line 251, the revised statement in the discussion is “Due to similar calorie diets, exercise recommendations and three major nutrients ratios (55% carbohydrate, 30% fat, and 15% protein) between the two groups. The WSP-MR could provide 15.5 g fiber and average glycemic index was 36.2 that may causes slow down energy absorption in subject. The Low-GI foods may enhance weight control because they promote satiety, minimize postprandial insulin secretion, and maintain insulin sensitivity. This hypothesis is supported by several intervention studies in humans in which energy-restricted diets based on low-GI foods produced greater weight loss than did equivalent diets based on high-GI foods [1].
Point 3: The manuscript cites the role of polyphenols and it might be worth citing references below where similar results, i.e. significant improvement of sub-optimal properties in sub-normal individuals (e.g. overweight), were observed
1. Yoshino, J.; Conte, C.; Fontana, L.; Mittendorfer, B.; Imai, S.; Schechtman, K.B.; Gu, C.; Kunz, I.; Rossi Fanelli, F.; Patterson, B.W.; Klein, S. Resveratrol supplementation does not improve metabolic function in nonobese women with normal glucose tolerance. Cell Metab. 2012, 16, 658-64, doi: 10.1016/jNaNet.2012.09.015.
2. Ellis, T.P.; Wright, A.G.; Clifton, P.M.; Ilag, L.L. Postprandial insulin and glucose levels are reduced in healthy subjects when a standardised breakfast meal is supplemented with a filtered sugarcane molasses concentrate. Eur J Nutr. 2016, 55, 2365-2376, doi:10.1007/s00394-015-1043-6.
Response 3: In line 311, the revised statement in the discussion is “Yoshino et al. reported that after 12 weeks of supplementation with resveratrol (75 mg/day), there was no decrease in body weight for non-obese women with normal glucose tolerance”[3]. A phytochemical and mineral-rich filtered sugarcane molasses concentrate (FMC) as an agent that can reduce insulin responses and lessen the load on the pancreatic beta cells [4].
Point 4:Minor edits, In the abstract, it stated that there were 56 subjects but the results section shows a total of 58.
Response 4: The correct subjects (n=58) were showed in revised abstract (line 20).
Point 5:Table 5 - formatting of the data in the first 3 rows (FBG, HbA1c and insulin) - alignment of rows
Response 5: The first 3 rows (FBG, HbA1c and insulin) in Table 5 were formatted.
References:
1. Röhrig, B.; du Prel, J.-B.; Wachtlin, D.; Kwiecien, R.; Blettner, M. Sample size calculation in clinical trials: part 13 of a series on evaluation of scientific publications. Deutsches Arzteblatt international 2010, 107, 552-556, doi:10.3238/arztebl.2010.0552.
2. Brand-Miller, J.C.; Holt, S.H.A.; Pawlak, D.B.; McMillan, J. Glycemic index and obesity. The American Journal of Clinical Nutrition 2002, 76, 281S-285S, doi:10.1093/ajcn/76/1.281S.
3. Yoshino, J.; Conte, C.; Fontana, L.; Mittendorfer, B.; Imai, S.; Schechtman, K.B.; Gu, C.; Kunz, I.; Rossi Fanelli, F.; Patterson, B.W.; Klein, S. Resveratrol supplementation does not improve metabolic function in nonobese women with normal glucose tolerance. Cell Metab. 2012, 16, 658-64, doi: 10.1016/jNaNet.2012.09.015.
4. Ellis, T.P.; Wright, A.G.; Clifton, P.M.; Ilag, L.L. Postprandial insulin and glucose levels are reduced in healthy subjects when a standardised breakfast meal is supplemented with a filtered sugarcane molasses concentrate. Eur J Nutr. 2016, 55, 2365-2376, doi:10.1007/s00394-015-1043-6.
Round 2
Reviewer 1 Report
The authors have done a good job of responding to the critiques and the paper will be a nice contribution to the literature.
Reviewer 2 Report
I think that you have answered appropriately to my questions.